# Impact of HIV co-infection on the evolution and transmission of multidrug-resistant tuberculosis

**Vegard Eldholm**[1]*[†], **Adrien Rieux**[2]†‡, **Johana Monteserin**[3,4], **Julia Montana Lopez**[2], **Domingo Palmero**[5], **Beatriz Lopez**[3,4], **Viviana Ritacco**[3,4], **Xavier Didelot**[6]*, **Francois Balloux**[2]*

[1]Division of Infectious Disease Control, Norwegian Institute of Public Health, Oslo, Norway; [2]UCL Genetics Institute, University College London, London, United Kingdom; [3]Instituto Nacional de Enfermedades Infecciosas, ANLIS Carlos Malbrán, Buenos Aires, Argentina; [4]Consejo Nacional de Investigaciones Científicas y Técnicas (CONICET), Buenos Aires, Argentina; [5]División Tisioneumonología, Hospital Muñiz, Buenos Aires, Argentina; [6]Department of Infectious Disease Epidemiology, Imperial College London, London, United Kingdom

**Abstract** The tuberculosis (TB) epidemic is fueled by a parallel Human Immunodeficiency Virus (HIV) epidemic, but it remains unclear to what extent the HIV epidemic has been a driver for drug resistance in *Mycobacterium tuberculosis (Mtb)*. Here we assess the impact of HIV co-infection on the emergence of resistance and transmission of *Mtb* in the largest outbreak of multidrug-resistant TB in South America to date. By combining Bayesian evolutionary analyses and the reconstruction of transmission networks utilizing a new model optimized for TB, we find that HIV co-infection does not significantly affect the transmissibility or the mutation rate of *Mtb* within patients and was not associated with increased emergence of resistance within patients. Our results indicate that the HIV epidemic serves as an amplifier of TB outbreaks by providing a reservoir of susceptible hosts, but that HIV co-infection is not a direct driver for the emergence and transmission of resistant strains.

*For correspondence: v.eldholm@ gmail.com (VE); xavier.didelot@ gmail.com (XD); f.balloux@ucl.ac. uk (FB)

†These authors contributed equally to this work

Present address: ‡CIRAD, UMR PVBMT, La Réunion, France

Competing interests: The authors declare that no competing interests exist.

## Introduction

Among the estimated 1.5 million people who died from TB in 2013, 360,000 were HIV co-infected and 200,000 cases were caused by multidrug-resistant TB (MDR-TB) (*World Health Organization, 2015*). Until the late 1980s, reports of MDR-TB were rare, and transmission of such strains was even less frequent (*Reves et al., 1981*; *Small et al., 1993*; *Wells et al., 2007*). The MDR-TB burden surged concurrently with the human immunodeficiency virus (HIV) pandemic and most reported early MDR-TB outbreaks mainly affected HIV co-infected individuals in hospitals and prisons (*Small et al., 1993*; *Wells et al., 2007*; *Ritacco et al., 1997*).

There are good epidemiological reasons to suspect that the HIV and MDR-TB pandemics are fueling each other. Not only does HIV infection render people more susceptible to develop active TB by weakening their immune system, but anti-TB drugs can also directly interfere with antiretroviral treatment. Rifampicin (RIF), one of the cornerstones in anti-TB therapy, has been shown to significantly lower serum concentrations of HIV protease and reverse transcriptase inhibitors (*Burman et al., 1999*; *Centers for Disease Control and Prevention, 1998*). To make matters worse, HIV co-infection is also associated with malabsorption of anti-TB drugs. This pattern is particularly pronounced for RIF, but seems to hold true for most anti-TB drugs (*Patel et al., 1995*; *Peloquin et al., 1993*).

**eLife digest** Tuberculosis is an infectious disease caused by a bacterium called *Mycobacterium tuberculosis* that causes more deaths worldwide than any other infection. Individuals who are infected with the Human Immunodeficiency Virus (HIV), which weakens the immune system, are particularly vulnerable to tuberculosis. However, treating individuals who are infected with both HIV and tuberculosis is complicated because the drugs currently used to treat one infection can interfere with the effectiveness of the drugs used to treat the other.

Tuberculosis is generally treated with antibiotics. However, some strains of *M. tuberculosis* are difficult to treat as they have evolved to resist the effects of multiple types of antibiotics. These "multidrug-resistant" bacteria appear to be particularly common in areas where HIV infections are also common. However, it was not known whether HIV directly influences whether *M. tuberculosis* bacteriaevolve into drug-resistant forms.

Eldholm, Rieux et al. have now analyzed the genomes, or total genetic content, of 252 samples of *M. tuberculosis* taken from the largest outbreak to date of multidrug-resistant tuberculosis in South America. This made it possible to identify the genetic mutations that enable the bacteria to resist antibiotic treatment. Using mathematical models to reconstruct the spread of multidrug resistant *M. tuberculosis* bacteria during the outbreak also made it possible to assess who transmitted tuberculosis to whom.

The results suggest that *M. tuberculosis* does not evolve drug resistance any faster in patients with HIV than otherwise. Furthermore, patients infected with both HIV and tuberculosis did not transmit tuberculosis to others more often than patients who did not have HIV. However, being infected with HIV did increase the likelihood that an individual would contract tuberculosis. HIV also increased the rate at which the symptoms of tuberculosis progressed in an individual.

To clarify the effect of HIV on the spread of tuberculosis, similar studies are needed that collect more complete patient data, including their anti-HIV treatment history and their degree of immune weakening.

HIV co-infection might also directly contribute to the accumulation of resistance in *Mtb*. First, as resistance mutations generally entail a fitness cost to the bacterium (at least initially), some resistant strains might be more successful in HIV+ hosts with weakened immunity leading to a reduced selective pressure on the bacillus. Second, some antiretroviral drugs used to treat HIV might have a mutagenic effect on mycobacterial genomes, but this has yet to be investigated (*McGrath et al., 2014*).

HIV co-infection and very low CD4 lymphocyte counts (<100 cells/mm$^3$), a hallmark of advanced HIV infection, have been shown to be risk factors for developing resistance to RIF and to a lesser degree isoniazid (INH) (*Bradford et al., 1996*; *Burman et al., 2006*; *Li et al., 2005*; *Porco et al., 2013*). However, a systematic review of 32 studies assessing MDR-TB prevalence by HIV status did not demonstrate an overall association between acquired MDR-TB and HIV, but suggested that HIV co-infection is a risk factor for contracting primary MDR-TB (*Suchindran et al., 2009*). In summary, the association between HIV co-infection and *Mtb* drug resistance remains unclear, with a number of studies yielding conflicting results (*Small et al., 1993*; *Chum et al., 1996*; *Lukoye et al., 2011*; *Meyssonnier et al., 2012*; *Robert et al., 2003*). Attempts have also been made to model the impact of HIV on TB incidence and resistance (*Sergeev et al., 2012*), but *in lieu* of empirical data, such studies relied on a number of assumptions on both host and pathogen biology as well as the interactions between them.

It is beyond doubt that HIV has been a driver of increased TB incidence globally, but a recent review of the subject actually found HIV co-infection to be associated with decreased rates of TB transmission within households and between close contacts (*Kwan and Ernst, 2011*). This observation is possibly explained by differing manifestation of TB in HIV positives, namely less frequent cavitation and lower pulmonary bacillary load (*Kwan and Ernst, 2011*). External factors such as social isolation or HIV infected patients being followed up more closely than HIV negatives may also contribute to this pattern (*Kwan and Ernst, 2011*). Indeed, in a low-incidence setting of close follow-up, HIV co-infection was associated with reduced TB transmission (inferred by clustering) and TB among

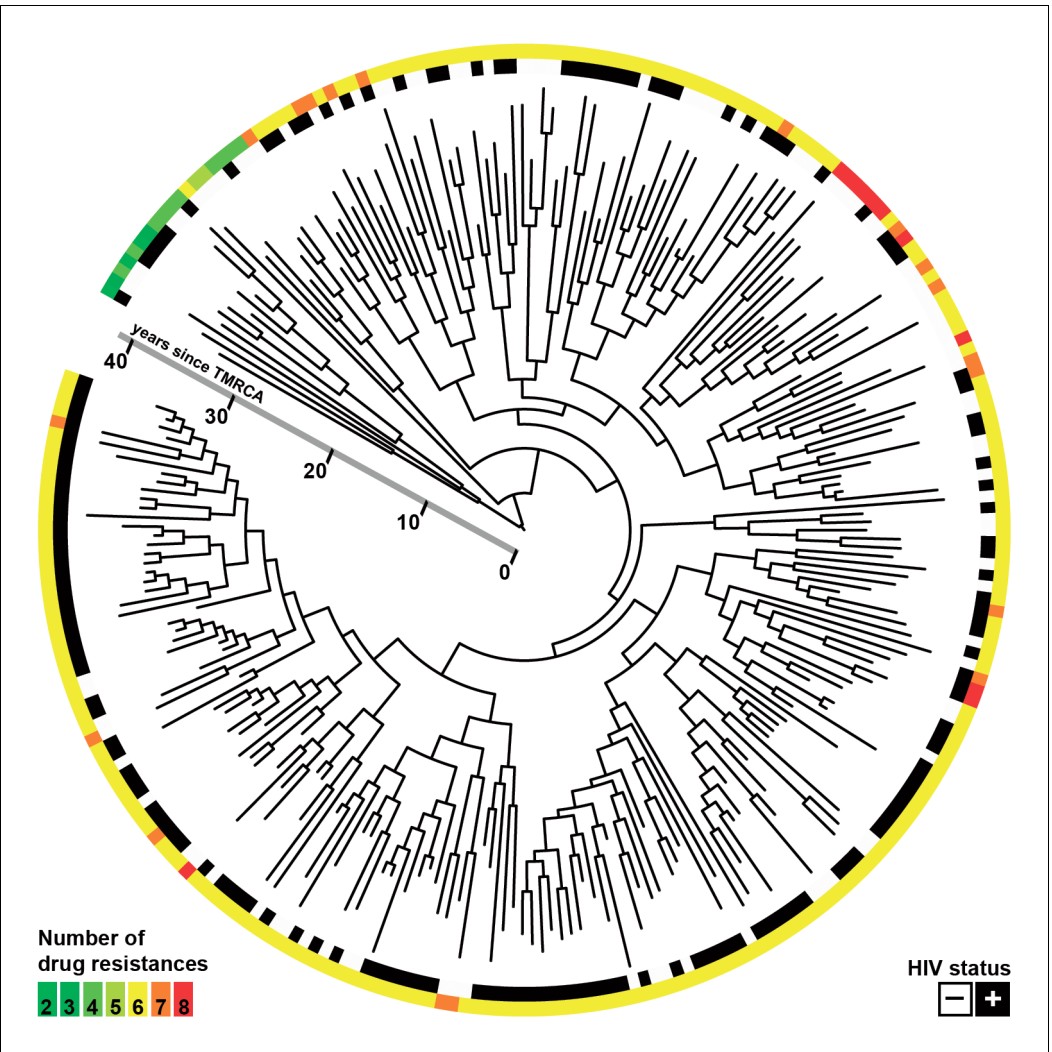

**Figure 1.** Whole-genome Bayesian evolutionary phylogeny of the M outbreak. The peripheral color strips indicate the HIV status of patients from which the clinical isolates were collected and the resistance burden of the isolate. The scale bar is given in years since the most recent common ancestor of the outbreak.

HIV co-infected was at least partly due to transmission from HIV-negative patients (*Fenner et al., 2012*).

In the current work we aimed to directly investigate the impact of HIV co-infection on the evolution of antibiotic resistance emergence and on transmission dynamics. We analyzed the genomes of 252 isolates belonging to the largest reported outbreak of MDR-TB in South America, caused by the M strain (*Ritacco et al., 1997*; *Eldholm et al., 2015*). The isolates were collected from patients with known HIV status from the mid-90s until 2009, providing important temporal information. To assess the impact of HIV co-infection on *Mtb* evolutionary rates, we estimated mutation rates in the terminal branches of a time-labelled phylogenetic tree, roughly corresponding to the evolutionary history of individual clinical *Mtb* isolates within sampled patients. We also inferred transmission networks by implementing a novel epidemiological model accounting for the long latency of TB. Finally, we estimated the length of the latent period by combining the results of the phylogenetic reconstruction and inferred transmission networks.

We found that HIV status of the host does not affect the mutation rate of *Mtb*, and that drug resistance is not more likely to evolve in HIV positive than HIV negative patients. Together these findings suggest that HIV co-infection is not a direct driver of *Mtb* drug resistance, which fits well

with the distribution of the global burden of TB, MDR-TB and HIV. Reconstructed transmission networks did not reveal a significant impact of HIV co-infection on the ability of patients to transmit TB. However, our estimates of TB latency confirm that HIV co-infection accelerates progression to active TB.

## Results

### Impact of HIV co-infection on Mtb mutation rates and resistance development

After filtering out positions with low mapping quality and removal of single nucleotide polymorphisms (SNPs) in problematic regions, a total of 509 SNPs separating the 252 isolates were used to construct a Bayesian phylogeny (*Figure 1*) (*Eldholm et al., 2015*). The majority of the isolates in the study shared the same six mutations yielding resistance to INH, RIF, streptomycin, kanamycin, pyrazinamide and ethambutol (*Eldholm et al., 2015*). The bulk of resistance mutations evolving within the outbreak were thus made up of ethionamide (ETH) and fluoroquinolone (FLQ) resistance mutations. The HIV status was known for all patients in the study, of which 60.7% were HIV positive.

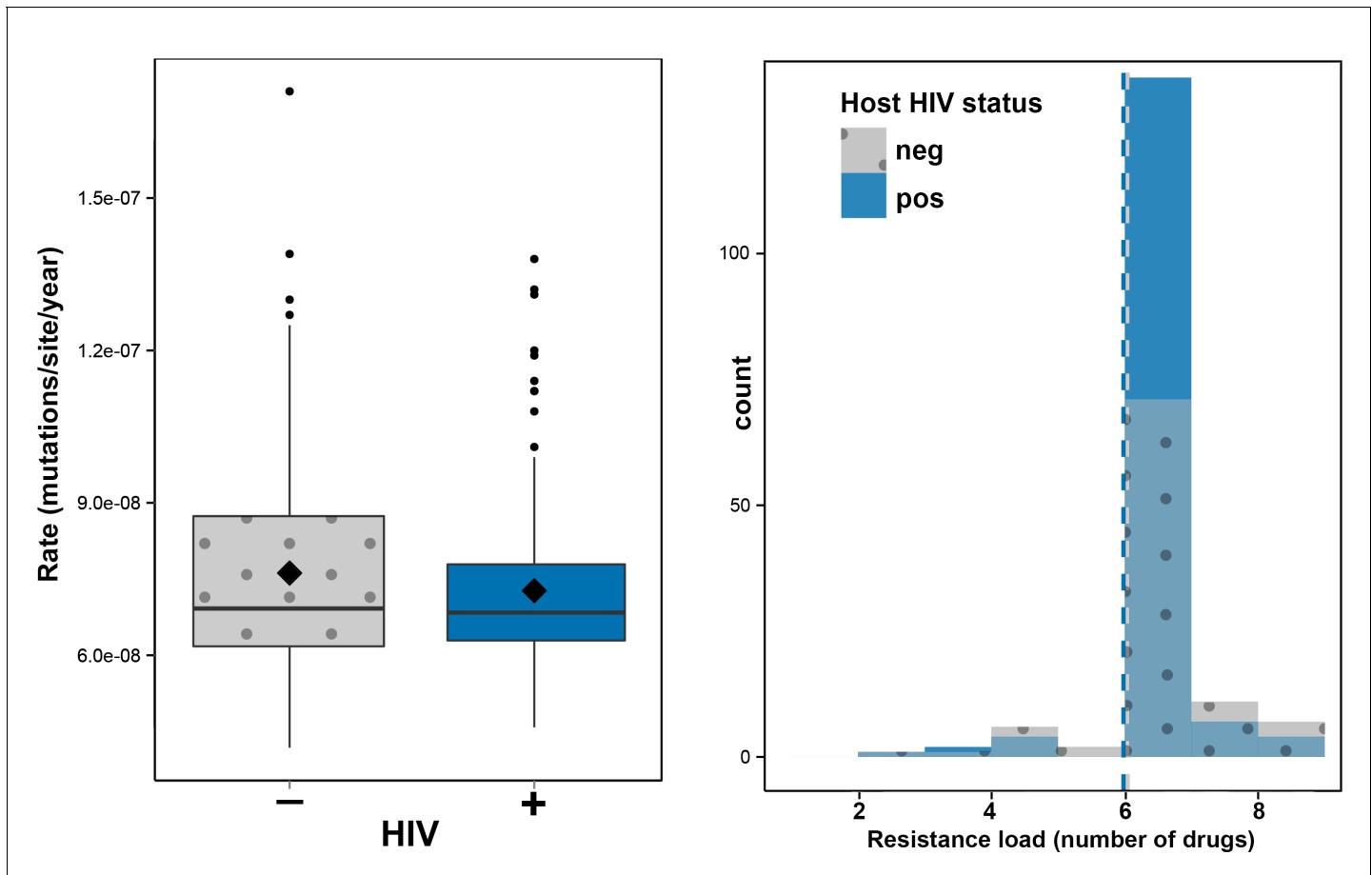

**Figure 2.** Impact of HIV co-infection on *Mtb* evolution. Left: Rate of evolution (substitutions/site/year) on terminal branches (p = 0.1920). Right: resistance load (number of antimicrobials to which resistance-conferring mutations were found in clinical *Mtb* isolates, stratified by HIV status of the host.

The following figure supplement is available for figure 2:

**Figure supplement 1.** Evolution of *Mtb* within patients as a function of HIV status.

**Table 1.** Number of SNPs accumulated in clinical isolates.

| Host HIV status | n | Mutations total | Mean number per isolate | $\chi^2$ p-value |
|---|---|---|---|---|
| Negative | 99 | 262 | 2.646 | < 0.001 |
| Positive | 153 | 277 | 1.810 | |

Based on the available data we considered that the sequenced outbreak isolates represented about one third of the total number of individuals belonging to the outbreak. Based on available RFLP data and estimates of the proportion of MDR-cases in Argentina belonging to the M strain, the outbreak is believed to have caused about 550 cases between 1992 and 2002, of which 109 genomes were available for study (20%). A large fraction of isolates from before 2001, which includes the peak of the outbreak, were lost in a freezer accident. From 2003 to 2009, the M strain caused 228 cases in Argentina, of which 143 genomes were available (63%), 116 isolates were sequenced from HIV positive patients and 85 from negatives. Lost isolates amounted to 40 positives and 25 negatives. For these years there are hence no reason to suspect any bias in the HIV status of the sampled patients ($\chi^2$; p = 0.53). Lost samples can potentially inflate the length of the terminal branches as they can result in missing internal nodes, but any inflation in branch length is thus expected to apply equally to branches leading to isolates sampled from HIV positive and negative patients.

To investigate the impact of HIV-TB coinfection on the accumulation of mutations in *Mtb* genomes we directly counted the mutations occurring on terminal branches by performing an ancestral reconstruction analysis in PAML (*Table 1*, *Figure 2—figure supplement 1*) (*Yang, 2007*). We observed no significant differences in the rate at which substitutions accumulate in the genomes of strains evolving in HIV positive and negative patients (*Figure 2a*). However, terminal branches were significantly longer and contained significantly higher numbers of mutations in HIV negative patients than in positives (*Table 1* and *Figure 2—figure supplement 1*), possibly reflecting a slower progression of TB in HIV negatives relative to positives.

Hypothesizing that HIV-coinfection could either be a direct driver of resistance emergence or increase the susceptibility to contract additionally resistant isolates, we tested whether patient HIV status was associated with resistance load, by counting the number of resistance determinants present in each *Mtb* isolate (*Supplementary file 1*: Sample information) and stratifying the data by HIV status. We found that the resistance load was near identical between *Mtb* isolates from HIV positive and negative patients (mean = 5.99 and 6.04 respectively) (*Figures 1* and *2b*). These results are in line with those from a very recent study of drug resistant TB in Kwazulu-Natal (*Cohen et al., 2015*).

The analysis above does not distinguish between mutations that emerged in the patients included in our sample (acquired resistance) and those acquired earlier in unsampled patients and subsequently transmitted to the patients in our sample (primary resistance). To investigate the impact of HIV co-infection on evolution of new resistance, we collected available treatment history for patients from whom isolates with terminal branch resistance mutations had been sampled (*Supplementary file 2*: Treatment histories). We excluded secondary mutations in resistance genes, namely *katG* and *rpoB* mutations, in isolates already harboring high-level resistance mutations in these genes. These mutations could either be random events or be involved in fitness compensation, but not resistance per se. We then excluded isolates collected from patients who had not been treated with drugs relevant for the terminal branch resistance mutation, as these most likely represent mutations that evolved in unsampled cases and subsequently transmitted to a sampled secondary case. This left 13 resistance mutations that evolved with high probability during therapy in 11 patients (*Table 2*). Nine events of acquired resistance occurred in seven HIV negatives and four in HIV positives. Based on the frequency of HIV co-infection among the sampled patients, HIV negative patients were overrepresented among cases of acquired resistance, but the difference was not significant (p= 0.24, Fisher's exa ct test). While the sample size is arguably small, this finding does also not implicate HIV as a driver of Mtb drug resistance within the outbreak.

**Table 2.** Identified events of within-patient acquired resistance.

| Isolate ID | HIV | Treatment history | Mutation | Acquired resistance |
|---|---|---|---|---|
| 107 | - | follow-up (ETH* treated) | *ethA* L225fs | ETH |
| 108 | - | follow-up (ETH and FLQ treated) | *ethA* S208P | ETH |
| 516 | - | follow-up (unknown treatment) | *pncA* D129G | PZA |
| 1757 | - | follow-up (ETH and FLQ treated) | *ethA* H22P | ETH |
| 2098 | - | follow-up (ETH and FLQ treated) | *ethA* F302S + *gyrB* D461V | ETH + FLQ |
| 2485 | - | follow-up (unknown treatment) | *ethA* G437fs | ETH |
| POGU | - | follow-up (ETH and FLQ treated) | *ethA* R259fs + *gyrB* R292G | ETH + FLQ |
| 110 | + | follow-up (ETH and FLQ treated) | *gyrB* R446S | FLQ |
| 257 | + | follow-up (ETH and FLQ treated) | *inhA* -15 C>T | ETH |
| 1298 | + | follow-up (ETH and FLQ treated) | *gyrA* D94N | FLQ |
| 2569 | + | follow-up (ETH treated) | *ethA* S251fs | ETH |

*Patient received the ETH analogue prothionamide

## Effect of HIV co-infection on TB transmission

To investigate the impact of HIV co-infection on transmission of *Mtb*, we implemented a new method to infer transmission events based on the timed phylogenetic tree (*Figure 1*). This was needed because a phylogeny is not directly informative about transmission events as a result of within-host diversity and evolution (*Didelot et al., 2016*; *Pybus and Rambaut, 2009*). Our methodology is briefly outlined below and explained in more details in the materials and methods section. A coalescent within-host model (*Didelot et al., 2014*) was combined with a Susceptible-Exposed-Infectious-Removed (SEIR) epidemiological model (*Lekone and Finkenstädt, 2006*). The likelihood of transmission from one host to another can be computed under this combined model, and this calculation was performed for all pairs of individuals with one acting as potential infector and the other as potential infectee. The likelihood calculation relies solely on the dates at which the two individuals were sampled, their relative position on the phylogeny, and whether the putative infector was smear positive or negative. It does not incorporate other information such as HIV status, so that these effects can be tested separately.

The SEIR model was set up with parameters for latency (mean of 5 years with 95% CI 46 days – 18.5 years) and infectious period (mean of 120 days with 95% CI 3–443 days). The infectious period includes time from symptom onset to infection clearance. A standard method for diagnosing TB is direct microscopy of sputum smears. If bacteria are visible under the microscope, the case is denoted smear positive. If no bacteria are observed, but *Mtb* can be cultured from the sputum, the case is culture positive but smear negative. Smear positive cases transmit TB far more efficiently on average than smear negative cases. We thus applied a so-called smear-correction, penalizing transmission event likelihoods involving a smear negative transmitter by multiplying likelihood values with 0.05.

The resulting matrix contains likelihoods of all possible transmission events (*Figure 3—source data 1*). For each of the 252 sampled cases in the outbreak, we extracted the most likely transmitter, resulting in 251 identified transmission pairs. Examples of transmission graphs and transmission events mapped on the phylogeny are shown in *Figure 3* whereas full transmission graphs are presented as figure supplements (*Figure 3—figure supplements 1* and *2*). *Figure 3—source data 2* provides the links between transmission graph nodes and sample IDs. We performed a simulation analysis to test the accuracy of our transmission analysis method, and sensitivity analyses to ensure that our results were robust to parameter choice (see Materials and methods).

Next, we analyzed transmission events as a function of the HIV status of the transmitter of transmitter-receiver pairs. We found no significant effect of HIV status on the ability of patients to cause secondary TB cases (*Table 3*). Due to incomplete sampling, a proportion of identified transmission pairs are expected to be spurious, as unsampled intermediary hosts go undetected. To account for

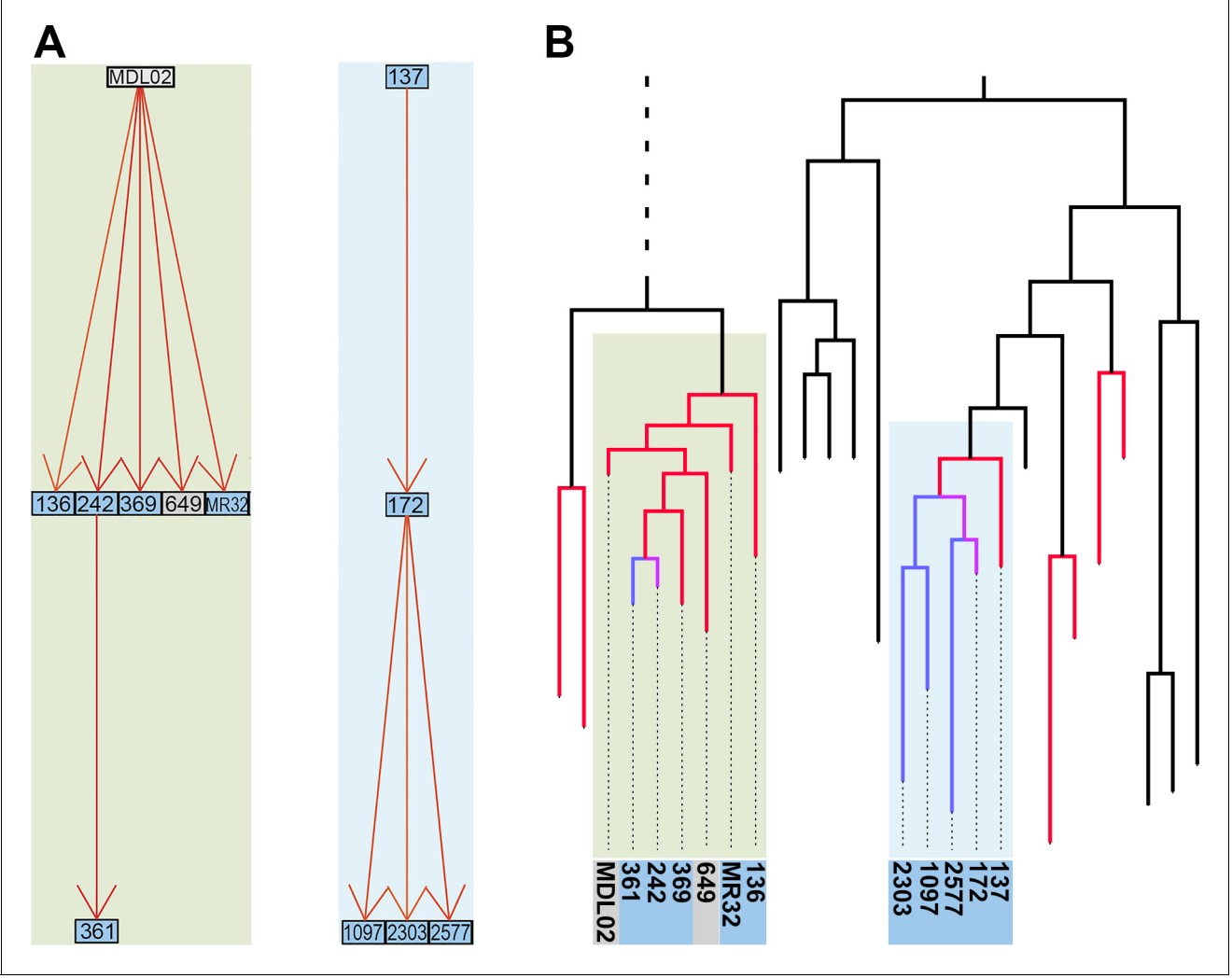

**Figure 3.** Reconstruction of transmission events. (**A**) Graphs representing two selected high-likelihood transmission chains. The colors of the edges indicate the probabilities of each transmission event from high (red) to lower (orange). Patient HIV-status is indicated by grey (negative) and blue (positive). (**B**) The corresponding transmission chains annotated in the timed phylogenetic tree. Red color highlights isolates linked by transmission events from a single source. Branches in magenta indicate subsequent transmission from a secondary case to additional cases (blue).

The following source data and figure supplements are available for figure 3:

**Source data 1.** Likelihood matrix of all possible pairwise transmission events.

**Source data 2.** Conversion table linking transmission graph nodes and sample IDs.

**Figure supplement 1.** Inferred transmission graph including all 251 transmission events (grey boxes HIV negative; blue HIV positive).Graph edges colored by likelihood from high (red) to low (yellow).

**Figure supplement 2.** Inferred transmission graph including only the most likely transmissions after applying various cut-offs (grey boxes HIV negative; blue HIV positive).Graph edges colored by likelihood from high (red) to low (yellow).

**Figure supplement 3.** Top 25% likely transmission events mapped on the timed phylogeny.

this, the analyses were repeated including only the most likely transmission events using three thresholds of increasing stringency (top 45%, 35% or 25% most likely transmissions). These subsets are expected to be increasingly enriched for true transmission pairs, but subsampling did not affect

**Table 3.** Number of reconstructed transmission events.

| Transmission event cut-off | Donor HIV status | Observed | Expected | Obs/Exp | $\chi^2$ p value |
|---|---|---|---|---|---|
| All transmissions | Negative | 80 | 98.61 | 0.81 | 0.3185 |
| | Positive | 171 | 152.39 | 1.12 | |
| Top 25% events | Negative | 20 | 24.75 | 0.81 | 0.2205 |
| | Positive | 43 | 38.25 | 1.12 | |
| Top 35% events | Negative | 30 | 34.57 | 0.87 | 0.3185 |
| | Positive | 58 | 53.43 | 1.09 | |
| Top 45% events | Negative | 36 | 44.39 | 0.81 | 0.1060 |
| | Positive | 77 | 68.61 | 1.12 | |

the original findings (**Table 3**). We also explicitly investigated the distribution of the number of transmissions per transmitter to test whether this could be affected by HIV status, but detected no significant differences between HIV-status of transmitters (**Table 4**). The 25% most likely infection events were mapped onto the time-labelled phylogeny for a visual integration of the modelled transmission links (**Figure 3—figure supplement 3**).

To further assess performance of the epidemiological modelling, we investigated whether six pairs of isolates with known epidemiological links (epi-pairs) had been identified by the transmission analysis. Four pairs of household contacts were identified as likely transmission pairs by the genomic analysis. All four were among the 35% most likely transmission events, and two among the top 25%. The SNP differences between these epi-pairs ranged from one to three SNPs (**Supplementary file**

**Table 4.** Distribution of transmissions as a function of HIV status of transmitter.

All transmission events

| HIV status | Transmissions per transmitter: | | | | | | | | | | | | Kruskal-Wallis p value |
|---|---|---|---|---|---|---|---|---|---|---|---|---|---|
| | none | 1 | 2 | 3 | 4 | 5 | 6 | 7 | 8 | 9 | 10 | 11 | |
| neg | 50 | 37 | 4 | 2 | 1 | 5 | 0 | 0 | 0 | 0 | 0 | 0 | 0.075 |
| pos | 63 | 58 | 11 | 10 | 5 | 2 | 2 | 1 | 0 | 0 | 0 | 1 | |

Top 25% likely transmission events

| HIV status | Transmissions per transmitter: | | | | | | |
|---|---|---|---|---|---|---|---|
| | none | 1 | 2 | 3 | 4 | 5 | |
| neg | 83 | 15 | 0 | 0 | 0 | 1 | 0.304 |
| pos | 121 | 25 | 3 | 4 | 0 | 0 | |

Top 35% likely transmission events

| HIV status | Transmissions per transmitter: | | | | | | |
|---|---|---|---|---|---|---|---|
| | none | 1 | 2 | 3 | 4 | 5 | |
| neg | 75 | 21 | 2 | 0 | 0 | 1 | 0.505 |
| pos | 111 | 33 | 4 | 4 | 0 | 1 | |

Top 45% likely transmission events

| HIV status | Transmissions per transmitter: | | | | | | |
|---|---|---|---|---|---|---|---|
| | none | 1 | 2 | 3 | 4 | 5 | |
| neg | 69 | 27 | 2 | 0 | 0 | 1 | 0.324 |
| pos | 100 | 39 | 7 | 5 | 1 | 1 | |

**3**: SNP distances between epi-pairs). The remaining two epi-pairs were not identified as likely transmission events. These included one pair of household contacts and one pair of isolates from the same patient taken 4.5 years apart. The genomic differences were nine and five SNPs respectively, which explains why the model did not identify these as likely transmission pairs. Interestingly, drug resistance had evolved in one of the epidemiologically linked isolates in both of these pairs, but in none of the four other pairs. We previously showed that a large number of mutations can hitchhike in the genetic background of resistance mutations sweeping to fixation and hypothesized that such selective sweeps could potentially confuse the reconstruction of transmission events (*Eldholm et al., 2014*). These two cases might well exemplify such a situation. However, it cannot be ruled out that the epi-links actually represent independent sources of infection (re-infection in the serially sampled patient).

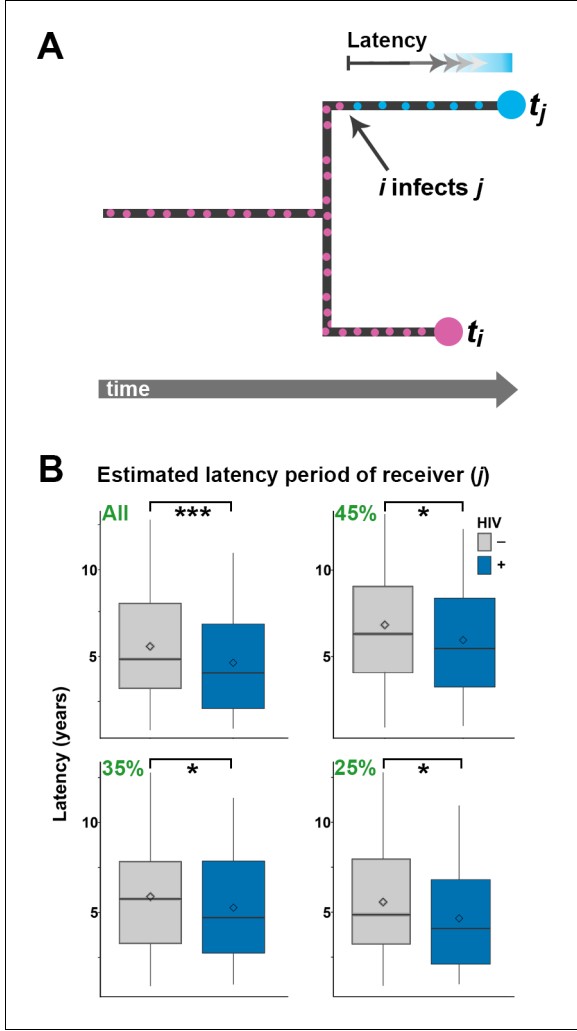

**Figure 4.** Estimating latency time as a function of HIV status. (A) For pairs of samples connected by a transmission event from *i* to *j*, transmission of *Mtb* is expected to have occurred on the terminal branch above *j*. Even though we do not know exactly when *j* went from latent TB to active TB, the latent period is included in the length of the terminal branch leading to *j* (see main text). We therefore use this branch length as an upwardly biased estimate for latency time. (B) For transmission pairs in the calculated transmission networks, the length (in years) of terminal branches leading to the recipient of the pairs (overestimated latency period) was extracted and stratified by HIV status of the recipient. To account for incomplete sampling, the analyses were performed on all 251 calculated transmission events as well as subsets including only the most likely transmission pairs (top 45, 35 and 25%). ***denotes p<0.001, *denotes p<0.05 as determined by unpaired t-test.

## Effect of HIV on progression of Mtb infection to active TB

We then set out to estimate the effect of HIV co-infection on the length of TB latency. For pairs of samples connected by a transmission event, transmission of *Mtb* from host A to B must happen after the date of the node connecting the two isolates in the Bayesian phylogeny (*Figure 1*) (*Didelot et al., 2012*, *2013*). The date of transition from silent infection to active TB is unknown, but must happen before sampling time, when the active status is known. An upward biased estimate for the length of latency period of individual *j* is therefore given by the difference between the date of the MRCA of the transmitter *i* and the receiver *j* (when *j* was not yet infected) and the date of sampling of *j* (by which time *j* had developed active TB). Although this estimator clearly overestimates the latency period, there is no *a priori* reason to suspect that the bias should be different between HIV negatives and positives. Any significant difference is therefore likely to reflect a difference in length of the actual latency period. Accordingly, we extracted the length (in years) of the branches separating the MRCA and recipient of the transmission pairs and stratified the data by HIV status of the recipient.

As we did not have an exhaustive sampling of all isolates in the outbreak, not all individuals would have donors present in the phylogenetic tree. To account for this, we analyzed branch lengths of the receiver for all 251 inferred transmissions, and separately for the 45%, 35% and 25% best supported transmission events, respectively. Again, we expected the proportion of genuine transmissions to increase in frequency as we restricted the analysis to a smaller subset of the best-supported transmissions. The length of the branches leading to HIV negative hosts was significantly longer than for HIV positive hosts when including all 251 estimated transmission events (p<0.001), and this difference remained significant for all three subsampling regimes (*Figure 4*).

When including only the top 25% of transmission events, the average branch lengths were 5.56 versus 4.65 years for HIV negative and positive receivers, respectively. A comprehensive review of 52 studies found that the average time from TB symptom onset to diagnosis (diagnostic delay) is approximately two months, with no significant difference between high and low income countries (*Sreeramareddy et al., 2009*). Another meta-study found that HIV positive status was associated with both increased and decreased diagnostic delay, depending on study setting (*Storla and Yimer, 2008*). The study most relevant to the current setting was conducted in 2005 in Buenos Aires and other Argentinean provinces and found a delay of about three months, with no significant effect of HIV status (*Zerbini et al., 2008*). As the difference in TB activation time we infer between HIV- and HIV+ is several times higher than the diagnostic delay reported in any setting, we feel confident that it reflects faster progression to active TB in HIV+ patients.

The fact that HIV co-infection significantly increases the rate of reactivation of latent TB is well documented. A comprehensive study from the United States found the rate of reactivation to be 25-fold higher in HIV co-infected individuals relative to their HIV-free peers (1.82 vs 0.072 per 100 person-year) (*Shea et al., 2014*). However, our outbreak analysis is necessarily restricted to people who develop active TB, and in this subset of cases, HIV co-infection seems to be associated with a relatively modest acceleration of TB progression, speeding up the process by about 11 months. We do not know when individual patients contracted TB and HIV respectively. Hypothetically, the accelerating effect of HIV co-infection on TB progression is likely to be underestimated in patients who were infected with HIV significantly later than TB. Conversely, patients who were infected with TB late in the study period might be enriched for HIV co-infection as these patients were more likely to develop TB in time for inclusion in the study. However, as the study period was relatively long, we do not believe this potential bias to have significantly affected our results.

## Discussion

The single most important impact of HIV infection in this large multi-decade outbreak of MDR-TB seems to be an increase in the proportion of patients who develop active TB. The HIV prevalence in Argentina is approximately 0.4% (in 2001 and 2014) (*World Health Organization, 2013*), whereas the proportion of HIV co-infected individuals is 60.7% within the M outbreak. These numbers demonstrate that HIV infection is a massive risk factor for developing TB with the MDR M strain. We found that HIV co-infection is associated with a moderately faster, yet statistically significant, progression to active TB. As this subtle effect of HIV status on time to active TB cannot explain the far higher incidence of the M strain in HIV positives, this suggests that the main effect of HIV co-

infection is to increase the absolute risk of developing active TB. In other words, we surmise that a large proportion of HIV negatives infected by the M strain will not progress to active TB but for those that do, the latency period is only slightly longer than for HIV positives.

This study encompasses an outbreak within which resistance to six common anti-TB drugs evolved early on, and our results are thus mainly restricted to the evolution of resistance to second-line drugs such as ETH and FLQ in individual isolates. Extrapolation of these findings to evolution of resistance to first-line drugs thus requires caution. However, the physiological and societal impact of HIV on TB

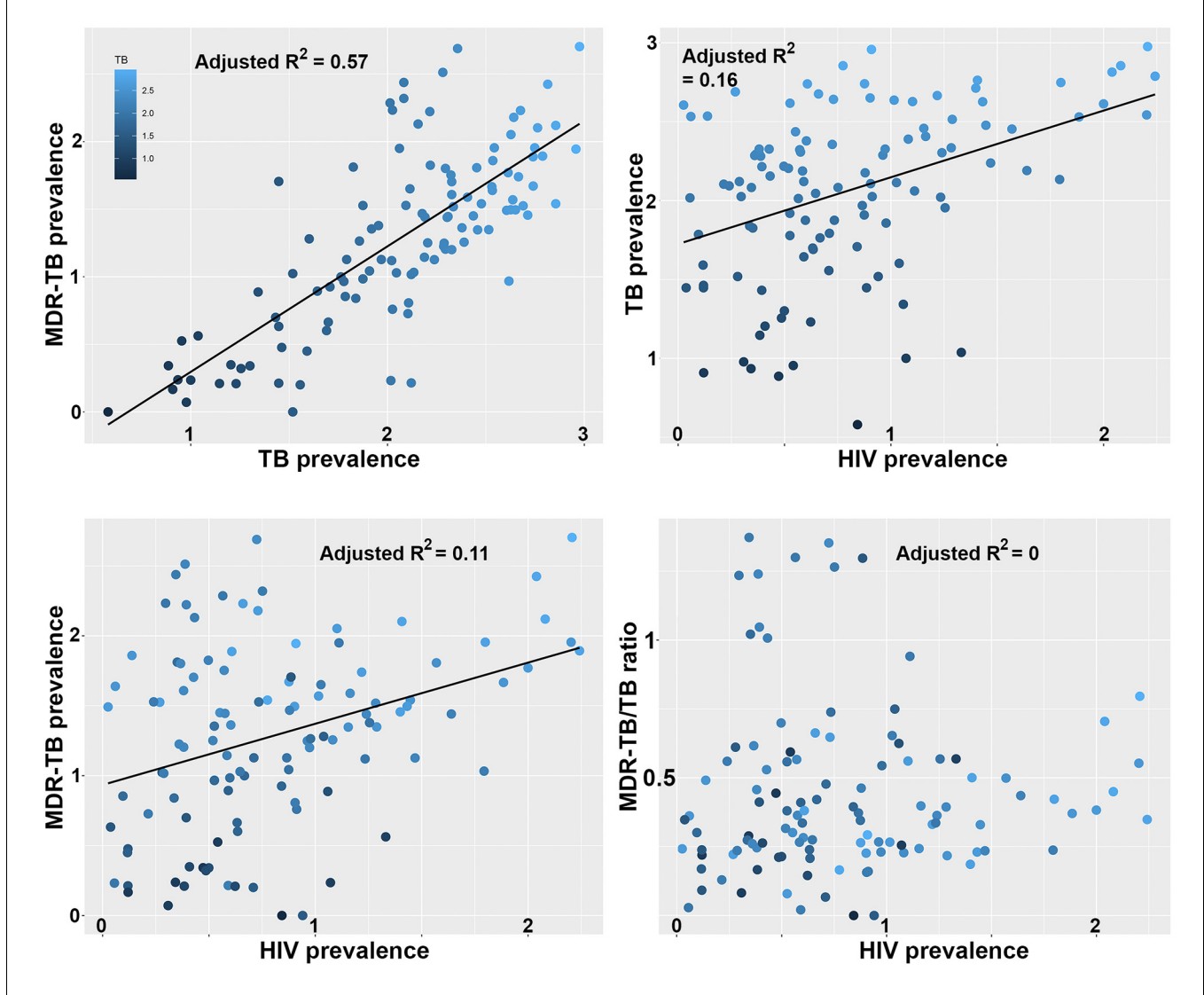

**Figure 5.** Correlations between global patterns of HIV, TB and MDR-TB prevalence. Clockwise: Per country prevalence of MDR-TB as a function of TB prevalence (p=2.2 × 10⁻¹⁶); TB prevalence as a function of HIV prevalence (p=5.9 × 10⁻⁶); MDR-TB prevalence as a function of HIV prevalence (p=1.6 × 10⁻⁴); Proportion of MDR-TB cases among TB patients as a function of HIV prevalence (p=0.8). All values are log-transformed. The depth of shading of individual dots reflect the TB prevalence in individual countries.

The following source data and figure supplement are available for figure 5:

**Source data 1.** Global per-country health, economy and disease metrics.

**Figure supplement 1.** Correlations between global patterns of HIV, TB and MDR-TB prevalence restricted to the top 50% countries in terms of GDP per capita.

patients as well as the fitness constraints associated with new *Mtb* resistance mutations should be fundamentally similar regardless of drug class.

It should be noted that free access to highly active antiretroviral therapy in Argentina from 1997 is likely to have mitigated the accelerating effect of HIV on TB progression (*Waisman, 2001*; *Gupta et al., 2015*). Clinical data on HIV progression was not available and we were thus unable to quantify the effect of anti-retroviral treatment by stratifying our analyses by CD4 counts. We predict CD4 counts would correlate negatively with TB progression. However, we do not expect that antiviral therapy (and increased CD4 counts) would nullify the effect of HIV on TB progression. Indeed, it has been shown that TB incidence during highly active anti-retroviral treatment is significantly higher than background levels even though a number of possible confounders makes the exact quantification of the effect of antiretroviral therapy challenging (*Gupta et al., 2015*; *Girardi et al., 2005*; *Lawn et al., 2005*; *Lawn and Wood, 2005*).

Accurate reconstruction of transmission chains of bacteria with extended periods of within-host evolution remains challenging (see Materials and methods). But even though some artefactual transmissions are bound to be included among the reconstructed high-confidence events, we are confident that the overall pattern of transmission is shaped by actual events and is hence robust. Restricting the transmission network analyses to the most likely transmission events did not affect our finding that HIV status does not significantly impact the transmissibility of *Mtb* (*Table 3* and *Table 4*). We also found that HIV co-infection does not affect the rate of *Mtb* evolution within patients. In fact, *Mtb* was found to accumulate more mutations in HIV negatives. This likely reflects the slower progression to active disease in this group, with these patients harboring *Mtb* for a more extended period relative to HIV positives. This pattern holds true also for antimicrobial resistance mutations, which were found to evolve significantly more often in HIV negatives than in HIV positives.

We previously showed that the largest clade in the M outbreak had evolved resistance to six antimicrobials by 1979, well before the HIV epidemic reached Argentina (*Eldholm et al., 2015*), a finding which has been replicated for another highly resistant *Mtb* lineage in South Africa (*Cohen et al., 2015*). To put our results in a global context, we retrieved data on the burden of TB, MDR-TB and HIV globally from the World Health Organization (WHO) Global Health Observatory Data Repository (http://apps.who.int/gho/data/node.main). We observed a strong correlation between TB and MDR-TB prevalence (*Figure 5a*) as well as a correlation between HIV and TB burden between countries (*Figure 5b*). We also recovered a highly significant correlation between HIV and MDR-TB (*Figure 5c*). However, when correcting the MDR-TB burden for total TB burden, the correlation vanished (*Figure 5d*). This is in line with our results on the M outbreak that HIV is a driver of TB in general, but does not disproportionately contribute to the rise of MDR-TB lineages.

By combining Bayesian evolutionary analyses and the reconstruction of transmission networks based on a new epidemiological model, we were able to directly assess the impact of HIV on the evolution and transmission of the single most widespread MDR-TB strain reported to date in South America. The main pre-extensively resistant (pre-XDR) clade within the outbreak evolved before the HIV epidemic in Argentina, but HIV patients at a major hospital in Buenos Aires played a central role in fueling the epidemic in the early 1990s (*Ritacco et al., 1997*; *Eldholm et al., 2015*), by providing the strain with a large and spatially restricted reservoir of individuals susceptible to develop active TB. Once the outbreak erupted, we find that HIV co-infection did not play a role in accelerating *Mtb* mutation rates; neither did HIV co-infected patients cause secondary TB cases at significantly higher rates than their HIV negative peers did. Our findings confirm that HIV co-infected patients have increased susceptibility to contract TB, but strongly suggest that they do not drive the evolution of *Mtb* resistance within an outbreak, nor do they act as super-spreaders of MDR-TB.

## Materials and methods

### Isolate collection

All available isolates belonging to the M outbreak as assessed by IS6110 RFLP were included in the study (see (*Eldholm et al., 2015*) for additional information on samples). The exact number of lost isolates is not known. No IS6110 RFLP data are available for isolates from before 1992; a freezer accident also contributed significantly to sample loss.

## Genomic analyses and phylogenetic evolutionary inferences

The protocols used for DNA isolation, preparation of sequencing libraries and SNP calling are described in (*Eldholm et al., 2015*), as are the methods for phylogenetic evolutionary inferences, testing of tip-based calibrations and molecular dating. Sequence reads from the study can be found under European Nucleotide Archive accession PRJEB7669. Briefly, BEAST 1.7.4 (*Drummond and Rambaut, 2007*) was used to infer a phylogeny, branch lengths and evolutionary rates using a general time reversible substitution model with variation among sites simulated using a discrete gamma distribution with four rate categories. We assumed a lognormal relaxed clock to allow variation in rates among branches in the trees. Trees were calibrated using tip dates only with sample time span ranging from October 1996 to December 2009. Following appropriate testing, we applied an exponential demographic model. Posterior distributions of parameters, including branch lengths and substitution rates were estimated by Markov chain Monte Carlo (MCMC) sampling.

## Analyzing differences in number of accumulated mutations between Mtb strains evolving in HIV-positive and negative patients

In this study, we aimed to test for evolutionary differences between strains evolving in HIV positive and negative patients. Because we can only be confident about the HIV status from which the samples were collected from, we restricted these analyses to terminal branches in the tree. We estimated the rates of evolution on terminal branches and compared those leading to HIV- and HIV+ hosts using two sample unpaired t-tests. We used the baseml model implemented in PAML program to perform the empirical Bayesian reconstruction of ancestral sequences. High-likelihood resistance mutations in the genes *embB, ethA, gidB, gyrA, gyrB, katG, ndh, mshA, pncA, rpoB, rpsL and rrs* were identified as described previously (*Eldholm et al., 2015*).

## Reconstruction of transmission chains and assessment of the impact of HIV co-infection

The code used to reconstruct transmission events is available at https://github.com/xavierdidelot/TransPairs. We wanted to reconstruct likely transmission events between sampled individuals, with the added difficulty that we knew a significant proportion of infected individuals were not sampled, so that some of the sampled individuals would have been infected by unsampled individuals. To avoid this difficulty, we developed the following inferential framework in which the likelihood of direct transmission from any sampled host to any other can be calculated. We consider a Susceptible-Exposed-Infectious-Removed (SEIR) model where individuals move from E to I at rate $\gamma_1$ and from I to R at rate $\gamma_2$. We also assume that within-host coalescence happens at a constant rate $\alpha$ as in previous work (*Didelot et al., 2014*). We want to calculate the likelihood $L_{i \rightarrow j}$ of transmission from host $i$ to host $j$ (*Figure 4*). Let $t_i$ and $t_j$ denote the known times at which the two hosts are sampled. Let $t_{i,j}$ denote the time at which the samples from $i$ and $j$ last shared a common ancestor, which is known from the timed phylogeny (*Figure 1*). Let $s$ denote the unknown time at which $i$ transmitted to $j$, assuming that this is indeed what happened. $s$ is unknown but is greater than $t_{i,j}$ and smaller than both $t_i$ and $t_j$. With these notations:

$$L_{i \longrightarrow j} = p(t_i, t_j, t_{i,j} \mid i \longrightarrow j) = \int p(t_i, t_j, t_{i,j} \mid s) p(s) \mathrm{d}s \quad \propto \int_{s=t_{i,j}}^{min(t_i, t_j)} p(t_i \mid s) p(t_j \mid s) p(t_{i,j}) \mid s) \mathrm{d}s$$

The first term in the integral is the probability of host $i$ being removed at time $t_i$ given that he was infectious at time s and is exponentially distributed with rate $\gamma_2$:

$$p(t_i \mid s) = \gamma_2 e^{-\gamma_2(t_i - s)}$$

The second term in the integral is the probability of host $j$ being removed at time $t_j$ given that he was exposed at time $s$ and so is a convolution of the exponentials with rates $\gamma_1$ and $\gamma_2$:

$$p(t_j \mid s) = \frac{\gamma_1 \gamma_2 (e^{-\gamma_2(t_j - s)} - e^{-\gamma_1(t_j - s)})}{\gamma_1 - \gamma_2}$$

The third term in the integral is the probability that coalescence of the two lines present in host $i$ at

time $s$ happens at time $t_{i,j}$ and also that either $i$ was infectious at time $t_{i,j}$ and stayed so until $s$ or that host $i$ was latent at time $t_{i,j}$ and became infectious (but not removed) by time $s$, leading to:

$$p(t_{i,j} \mid s) = \alpha e^{-\alpha(s-t_{i,j})} \frac{2\gamma_1 e^{-\gamma_2(s-t_{i,j})} - \gamma_2 e^{-\gamma_2(s-t_{i,j})} - \gamma_1 e^{-\gamma_1(s-t_{i,j})}}{\gamma_1 - \gamma_2}$$

By injecting the last three equations into the first we get the likelihood of transmission from $i$ to $j$. These calculations were made for all putative infector-infectee pairs using $\gamma_1 = 0.2$ per year and $\gamma_2 = 3$ per year and the previously estimated within-host coalescent rate $\alpha = 0.83$ per year (*Didelot et al., 2014*). The likelihoods of transmission from smear negative individuals was multiplied by 0.05 to reflect the lower infectiousness of these individuals. The SEIR epidemiological model assumed in the calculations above implies that there is random mixing between the individuals, with every infectious individual being *a priori* equally likely to infect any susceptible individual. Although the assumption of random mixing is appealing in theory, in practice human population are well known to behave differently, with for example a strong effect of the household structure in the trans- mission patterns of many pathogens (*Cauchemez et al., 2004*; *Whittles and Didelot, 2016*). Here

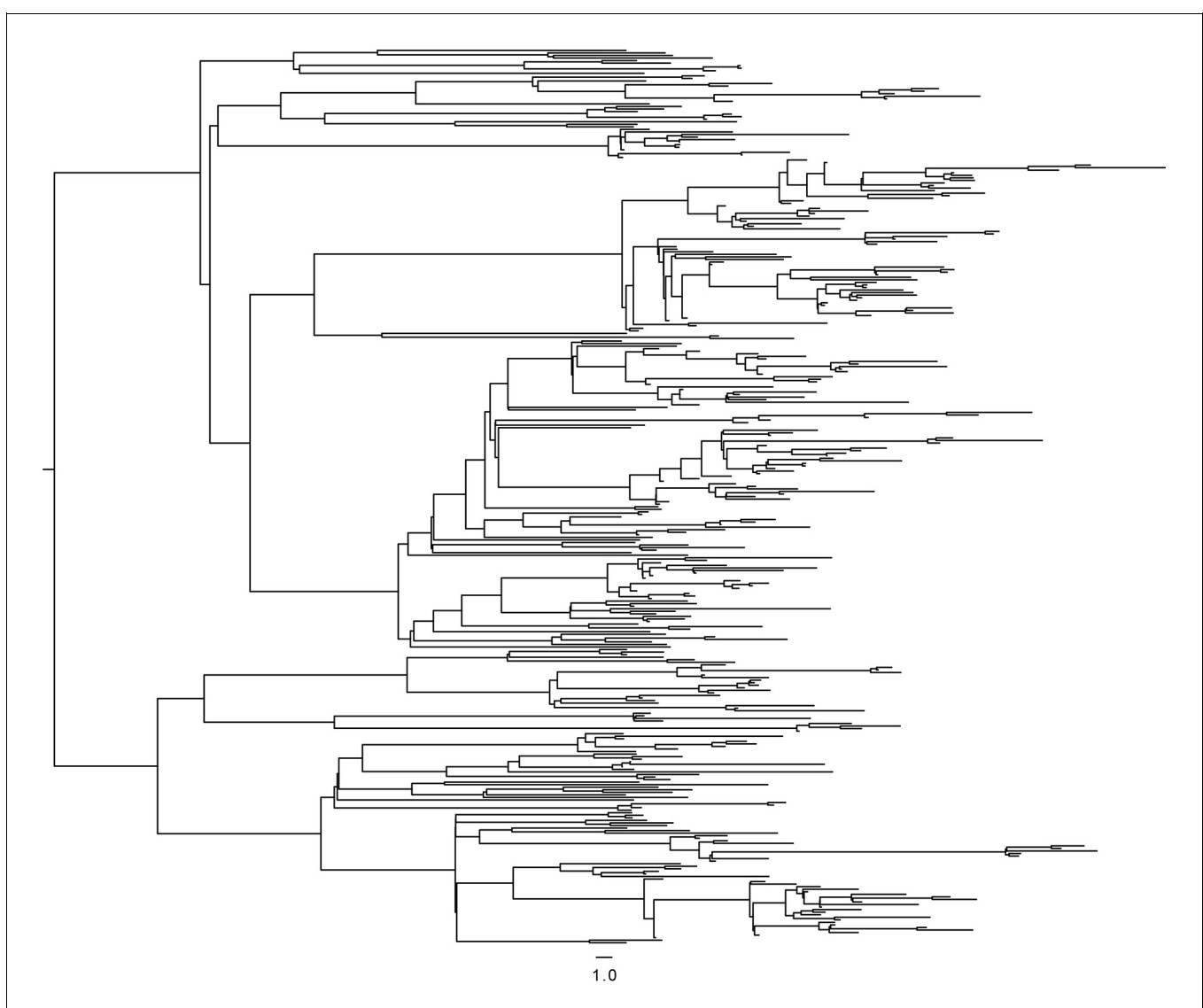

1.0

**Figure 6.** Timed phylogeny used in simulation of SEIR model.

we did not have information on the structure of the population and so could not integrate it in our model. Application of our methodology in a setting where such information is available could be performed simply by multiplying the likelihood values with the *a priori* probabilities of transmission caused by the host population structure.

From the full matrix of transmission likelihoods between all pair of strains, we aimed to reconstruct disease transmission as accurately as possible. For each pair [*i,j*] of the transmission matrix, we started by removing the lowest likelihood value (*i* infecting *j* or *j* infecting *i*). From the remaining transmission events, we used Edmonds algorithm implemented in the RBGL R package (*Carey et al., 2016*) to find the spanning arborescence of minimum weight (sometimes called an optimum branching). An optimum branching is a graph defined as a set of directed edges that contain no cycles and such that no two edges are directed towards the same node. In our reconstruction, such a graph contains *n* nodes, *n* being the number of isolates and *n-1* directed edges representing the transmission events. As our sampling of the outbreak was not exhaustive, we know that a proportion of direct transmission events did not happen. To deal with that situation, we used various thresholds of inferred transmission events with the highest associated likelihoods to plot the transmission graphs and analyze the distribution of transmission events.

## Simulations and sensitivity analyses

In order to test the accuracy of the above method of reconstruction of transmission chains, we simulated an SEIR model for a population of $N$ = 3000 individuals, with a transmission rate of $\beta$ = 0.001 per year, a rate of becoming infectious when exposed $\gamma_1$ = 0.2 per year, and a rate of being removed when infectious $\gamma_2$ = 3 per year. These values of $N$ and $\beta$ were selected to produce simulated outbreaks of roughly the same size as in the real data, and these values are not used for inference. The transmission tree generated by this simulation was recorded. A timed phylogeny was then constructed from the transmission tree, using a coalescent within-host evolutionary model with coalescent rate $\alpha$ = 0.83 per year, and leaves were randomly removed from this tree to simulate incomplete sampling of cases, keeping two thirds of the leaves in the second half of the outbreak to emulate the sampling frame in our study. The resulting phylogeny (*Figure 6*) was then analyzed in exactly the same way as the real data: the likelihood of transmission was computed for every pair of leaves, a transmission tree was deduced using Edmonds algorithm, and only the 25%, 35% or 45% most likely transmission links were retained to account for incomplete sampling. When applying these three thresholds to the simulated data, we found that the proportion of correctly inferred links were 74%, 69% and 63%, respectively. These results conform with our expectation given that there is significant uncertainty about who infected whom based on genomic data alone when accounting for extended periods of within-host evolution (*Didelot et al., 2016*; *Biek et al., 2015*; *Didelot et al., 2014*; *Worby et al., 2014*).

The results of our transmission analysis are based on three parameter values, namely a mean latent period of 5 years, a mean infectious period of 120 days and a smear correction of 0.05 by which the likelihood of transmission from smear negative individuals is multiplied. These parameters were selected based on the literature and clinical experience. The latent period can vary extensively between people. Approximately two months of diagnostic delay (*Sreeramareddy et al., 2009*) plus two months from treatment onset to clearance of the MDR infection (*Brust et al., 2013*) suggests that 120 days is a reasonable estimate of infectious period. Finally, A 20-fold decreased transmissibility of smear-negative cases was chosen as a reasonable parameter (*Ma et al., 2015*). We performed a sensitivity analysis to test how reliable our results would be if any of these parameters were inaccurate. For each of the three parameters, we ran the analysis again considering double and half of their specified values above, and compared the reconstructed transmission links with those of the main analysis. In each case, the proportion of links identical with the main analysis was between 91% and 99%. We also performed an analysis in which no smear correction was applied and recovered 90.5% of the links in the main analysis.

## Collection of data on global HIV, TB and MDR-TB burden

HIV prevalence expressed as% population between the age of of 15 and 49 was downloaded form the World Bank Data website (http://data.worldbank.org/indicator/SH.DYN.AIDS.ZS).

TB and MDR-TB prevalence data was obtained from the World Health Organization (http://www.who.int/tb/country/data/download/en/). For TB prevalence, data was available for all countries for the year 2013 and point estimates of prevalence by 100 k individuals were retrieved (e_prev_100 k).

For MDR-TB prevalence, the data was collected less systematically, and relies on a mix of surveillance, surveys and models. We used the estimated number of MDR-TB cases among all notified pulmonary TB cases (e_mdr_num), expressed as prevalence per 100 k individuals by dividing by country population size estimates from the same source. We calculated the proportion of MDR-TB cases by dividing the prevalence of MDR-TB by the prevalence of TB. All four variables (HIV-, TB-, MDR-TB-prevalence and the ratio of MDR-TB/TB prevalence were transformed as $\log(x+1)$ prior to analyses. Pearson correlation coefficients were used to test for significant associations between the prevalence of TB and MDR-TB, HIV and TB, HIV and MDR-TB and finally HIV and of the MDR-TB/TB ratio.

The robustness of the prevalence estimates likely vary between countries due to difference in methodology and surveillance effort, which may lead to some biases in the correlations reported in *Figure 5*. We reasoned that more robust estimates should be obtained in countries with more developed economies and public health institutions.

Thus, we additionally retrieved estimates for 2013 GDP per capita (http://data.worldbank.org/indicator/NY.GDP.PCAP.CD) and health expenditure (%) http://data.worldbank.org/indicator/SH.XPD.TOTL.ZS.

For all countries. Health expenditure was transformed into absolute health expenditure per capita, by multiplying by GDP and dividing by population size of the countries. The source data used in these analyses is provided in *Figure 5—source data 1*.

We then recomputed the correlations reported in *Figure 5* on different fractions (25%, 50% and 75%) of the countries with highest GDP or health expenditure per capita. Prevalence estimates from countries with lower GDP are indeed likely to be less robust as the coefficients between the significant correlations in *Figure 5* (panels A, B and C) were substantially higher for the countries with high GDP. However, importantly, we never recovered a significant correlation between the prevalence of HIV and the proportion of TB that were MDR-TB. In *Figure 5—figure supplement 1*, we report the correlations between the same variables than in *Figure 5* for the 50% countries with highest GDP.

## Acknowledgements

At the time of submission, the corresponding authors were unable to reach Julia Montana Lopez (co-author) and were as such unable to obtain a final approval from her.

## Additional information

### Funding

| Funder | Grant reference number | Author |
| --- | --- | --- |
| European Research Council | 260801-BIG_IDEA | Francois Balloux |
| Norges Forskningsråd | 221562 | Vegard Eldholm |
| National Institute for Health Research | HPRU-2012-10080 | Xavier Didelot |
| National Institute for Health Research | UCL Hospitals Biomedical Research Centre | Francois Balloux |

The funders had no role in study design, data collection and interpretation, or the decision to submit the work for publication.

### Author contributions

VE, FB, Conception and design, Analysis and interpretation of data, Drafting or revising the article; AR, VR, XD, Analysis and interpretation of data, Drafting or revising the article; JM, DP, BL, Acquisition of data, Drafting or revising the article; JML, Acquisition of data, Analysis and interpretation of data

## Author ORCIDs

Vegard Eldholm, http://orcid.org/0000-0001-6721-3375

Xavier Didelot, http://orcid.org/0000-0003-1885-500X

## Additional files

### Supplementary files

• Supplementary file 1. *M. tuberculosis* sample information.

• Supplementary file 2. Patient treatment histories.

• Supplementary file 3. SNP distances and transmission reconstruction results for samples pairs with known epidemiological link.

### Major datasets

The following previously published dataset was used:

| Author(s) | Year | Dataset title | Dataset URL | Database, license, and accessibility information |
|---|---|---|---|---|
| Eldholm V, Monte-serin J, Rieux A, Lopez B, Sobko-wiak B, Ritacco V, Balloux F | 2015 | Whole genome analysis of time-structured samples from an outbreak of Mycobacterium tuberculosis M strain in Buenos Aires | http://www.ebi.ac.uk/ena/data/view/PRJEB7669 | Publicly available at the EBI European Nucleotide Archive (accession no: PRJEB7669) |

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
