## [Decision Letter]

Thank you for submitting your article "Impact of HIV co-infection on the evolution and transmission of multidrug-resistant tuberculosis" for consideration by *eLife*. Your article has been reviewed by two peer reviewers, and the evaluation has been overseen by a Reviewing Editor and Prabhat Jha as the Senior Editor. The reviewers have opted to remain anonymous.

The reviewers have discussed the reviews with one another and the Reviewing Editor has drafted this decision to help you prepare a revised submission.

Summary:

The reviewers and Reviewing Editor were satisfied with the general direction of the analysis and found the work to be sophisticated and to hold an important public health message. However, there were a number of issues around the model and its parameterisation that we felt were inadequate, and which would need to be addressed for the manuscript to be considered further.

Essential revisions:

1) In the first paragraph of the subsection “Effect of HIV co-infection on TB transmission”, you appear to describe a new method for inferring transmission pairs (i.e. person A infected person B), which you have developed. You use a simple SEIR transmission model in determining who infected whom and use an SEIR simulation to help validate the method.

It is important that you further explain this method so that readers can better understand it and researchers will be better able to apply it judiciously in their own work. For example, you conducted a sensitivity analysis on the values used for the latent period and the removal rate but you did not include HIV status in the model, and HIV status could affect each of these parameters. Furthermore, you do not state what mixing assumption you used and likely assumed random mixing. It would be useful if you could explore how the method would perform in the context of non-random mixing (e.g. discuss what the expected effects would be and not necessarily conduct new simulations). You also reported that keeping two-thirds of cases in the second half of the simulated outbreak corresponded to the sampling frame used for the actual genomic data analyses but you need to explain where this number came from (subsection “Simulations and Sensitivity analyses”). If necessary, you should also test the impact of different sampling fractions.

You state "These results conform with our expectation given that there is significant uncertainty about who infected whom based on genomic data alone when accounting for extended periods of within-host evolution" but this high degree of uncertainty does not appear in the conclusions in the main text.

2) In the third paragraph of the subsection “Effect of HIV co-infection on TB transmission”, you tested whether HIV-positive or HIV-negative individuals transmit TB more often conditional on having TB disease. As you suggest, the analysis including all transmissions seems like it would not be very informative since it likely includes many pairs who did not actually transmit between them. While the top X% analyses are affected by the same issue to a lesser extent, there could be another bias in these analyses since enriching for actual transmission pairs would seem to condition on there being at least 1 transmission from the putative source case and thereby eliminates (many of) the instances in which no TB transmission occurred. This would skew the distribution of onward transmissions which could make HIV-positives and negatives appear more similar in terms of their transmission potential than is actually the case.

To provide a simple numerical example of the potential issue, suppose that out of 100 HIV-negative people with active TB 60 of them transmit to 1 other person, while 80 out of 100 TB-HIV coinfected people each transmit TB to 1 person. In this scenario, TB would appear to have the same transmissibility among those who transmitted (i.e. 1 TB transmission per TB transmitter in each HIV group). However, this would give a misleading picture of the actual TB transmission potential of HIV-positive vs. HIV-negative individuals since coinfected people had an 80% probability of transmitting whereas HIV-negative people had a 60% chance of transmitting.

3) In the last paragraph of the subsection “Effect of HIV on progression of *Mtb* infection to active TB”, you correlate cross-sectional data on HIV and TB prevalences and HIV and MDR-TB prevalences across countries in an effort to determine whether HIV increases TB in general or if it specifically drives the spread of MDR-TB. However, you should explain why this analysis is appropriate in light of the findings of Sergeev et al. (Sci Trans Med 2013, reference 20) who find that the "individual-level association between HIV and drug-resistant forms of TB is dynamic, and therefore, cross-sectional studies that do not report a positive individual-level association will not provide assurance that HIV does not exacerbate the burden of resistant TB in the community."

Also, while the analysis includes HIV and TB/MDR-TB, it's not clear on how it specifically fits in with the rest of paper or if it could be cut without loss of any key information.

You report in Figure 5, the correlation between global patterns of MDR-TB, TB and HIV prevalence. In the Methods you refer to the World Health Organization for the underlying data. This reference should be extended as it now only states "World Health Organization". Data from several countries are known to have limitations in the sense that the prevalence of HIV or TB may not always been known or collected in a rigorous manner. The prevalence of a particular disease can also vary within a single country. Use of these data can therefore also result in biased estimates, as estimates for HIV and TB may be derived from different sites in a single country.

This entire analysis needs to be either strengthened or simply removed from the paper.

4) More explanation is required for key numbers reported in their paper.

In particular, in the Results, you state: "Based on the available data we considered that the sequenced outbreak isolates represented about 35% of the total number of individuals belonging to the outbreak." You should explain how you came up with the figure of 35%. What are you assuming about the fraction of TB cases that are diagnosed and recorded? How, and over what time scale, are you defining the "outbreak" since reactivation TB cases may have been infected decades prior to disease onset?

Again, in the Results, you state: "This left 13 resistance mutations that evolved with high probability during therapy in 11 patients (Table 2). Seven of the patients were HIV negative and four were positive. A χ2 analysis revealed a statistically significant overrepresentation of acquired resistance in HIV-negative cases relative to positives (p = 0.027). This finding strongly suggests that HIV was not a driver of *Mtb* drug resistance within the outbreak." Given that you indicate that these results strongly back up a key part of your findings, it is important that more information be provided. Specifically, it is unclear exactly which numbers were being compared in the test (e.g. was the null hypothesis that there should be a 50:50 split between HIV-positives and negatives, or was it directly comparing the fraction of mutations found in HIV-infected vs. uninfected individuals [though this would not seem to give the p-value reported]). Also, a Fisher's exact test, not a chi-squared test, would likely be required given that apparently some of the cell sizes were very small.

---

## [Author Response]

*Essential revisions:*

*1) In the first paragraph of the subsection “Effect of HIV co-infection on TB transmission”, you appear to describe a new method for inferring transmission pairs (i.e. person A infected person B), which you have developed. You use a simple SEIR transmission model in determining who infected whom and use an SEIR simulation to help validate the method.*

It is important that you further explain this method so that readers can better understand it and researchers will be better able to apply it judiciously in their own work. For example, you conducted a sensitivity analysis on the values used for the latent period and the removal rate but you did not include HIV status in the model, and HIV status could affect each of these parameters. Furthermore, you do not state what mixing assumption you used and likely assumed random mixing. It would be useful if you could explore how the method would perform in the context of non-random mixing (e.g. discuss what the expected effects would be and not necessarily conduct new simulations). You also reported that keeping two-thirds of cases in the second half of the simulated outbreak corresponded to the sampling frame used for the actual genomic data analyses but you need to explain where this number came from (subsection “Simulations and Sensitivity analyses”). If necessary, you should also test the impact of different sampling fractions.

We have expanded our description of the methodology used to estimate transmission probabilities, both in the Results and Methods section. For a given putative infector-infectee pair, the calculation depends only on the dates of detection of the two cases, their relative position on the phylogeny, and whether the infector was smear positive or negative. It does not rely on HIV status, so that the effect of HIV status on transmission can be tested independently. We have clarified this important point in the Results section (subsection “Effect of HIV co-infection on TB transmission”, first paragraph). The underlying epidemiological model is a SEIR model, which assumes random mixing of the population, and we have clarified this in the Methods section (subsection “Reconstruction of transmission chains and assessment of the impact of HIV co-infection”, first paragraph) and now discuss how information on the host population structure could be integrated into our inferential framework. The code used to reconstruct transmission events is now available on GitHub https://github.com/xavierdidelot/TransPairs and the paper now includes a pointer to this website (in the aforementioned paragraph, the code should be available now or within a few days). The two-thirds sampling in the latter half of the outbreak reflects the fact that we know the completeness of sampling to be 63% (143/228) in the years 2003 – 2009 (2009 being the last year covered by the study). We have clarified the estimated sample loss in the early and late stages of sampling in the last paragraph of the subsection “Impact of HIV co-infection on *Mtb* mutation rates and resistance development”.

You state "These results conform with our expectation given that there is significant uncertainty about who infected whom based on genomic data alone when accounting for extended periods of within-host evolution" but this high degree of uncertainty does not appear in the conclusions in the main text.

We agree that this uncertainty could have been acknowledged more clearly in our conclusions. We have now added some reservations regarding the accuracy of the reconstruction of individual transmission events in the Discussion section, fifth paragraph.

*2) In the third paragraph of the subsection “Effect of HIV co-infection on TB transmission”, you tested whether HIV-positive or HIV-negative individuals transmit TB more often conditional on having TB disease. As you suggest, the analysis including all transmissions seems like it would not be very informative since it likely includes many pairs who did not actually transmit between them. While the top X% analyses are affected by the same issue to a lesser extent, there could be another bias in these analyses since enriching for actual transmission pairs would seem to condition on there being at least 1 transmission from the putative source case and thereby eliminates (many of) the instances in which no TB transmission occurred. This would skew the distribution of onward transmissions which could make HIV-positives and negatives appear more similar in terms of their transmission potential than is actually the case.*

To provide a simple numerical example of the potential issue, suppose that out of 100 HIV-negative people with active TB 60 of them transmit to 1 other person, while 80 out of 100 TB-HIV coinfected people each transmit TB to 1 person. In this scenario, TB would appear to have the same transmissibility among those who transmitted (i.e. 1 TB transmission per TB transmitter in each HIV group). However, this would give a misleading picture of the actual TB transmission potential of HIV-positive vs. HIV-negative individuals since coinfected people had an 80% probability of transmitting whereas HIV-negative people had a 60% chance of transmitting.

We appreciate the reviewers’ thorough assessment of our transmission reconstruction analyses. We agree that the distribution of transmission per potential transmitter is indeed worth a closer examination than we presented in the original submission. That said, we believe that the potential bias identified by the reviewers as illustrated by their numerical example is based on a misunderstanding of how these analyses were actually conducted. Based on the general depth and quality of the reviewers’ comments, this suggests to us we should explain our methodology better, as pointed out by the reviewers under point 1.

In the original submission we did not quantify the number of transmitters, but the number of reconstructed transmission events. We will illustrate this point with the same numerical example: 60% of HIV- and 80% of HIV+ transmitting to one secondary case and assuming we can perfectly reconstruct all transmissions. If we had counted the number of transmissions per person who have transmitted, we would indeed find 1 for both HIV+ and HIV-. However, what we counted were the number of transmissions and for the same numerical example we would thus have observed 4/3 more transmissions from HIV+ relative to HIV-.

This being said, we recognize that skewed distributions of transmissions, e.g. in the form of a limited number of transmitters responsible for a large portion of transmissions within one of the groups could indeed mask differences in transmissibility. We therefore extended our analyses to investigate the distribution of transmissions in more detail. For each subset of transmissions (all and top-X%) we counted the number of transmissions per transmitter (0, 1, 2, … n) and used non-parametric Kruskal-Wallis tests to assess whether the distributions were significantly different. When all transmission events were included, HIV-positive transmitters tended towards having transmitted more often than HIV-negatives, but the difference was not significant. After application of the various top-X% cut-offs the two groups became even more similar (Table 4). This suggests that even when taking into account the distribution of transmissions per transmitter, there is no major impact of HIV co-infection on transmissibility.

The extended results are included in Table 4 and discussed in the fourth paragraph of the subsection “Effect of HIV co-infection on TB transmission” in the revised manuscript.

3) In the last paragraph of the subsection “Effect of HIV on progression of Mtb infection to active TB”, you correlate cross-sectional data on HIV and TB prevalences and HIV and MDR-TB prevalences across countries in an effort to determine whether HIV increases TB in general or if it specifically drives the spread of MDR-TB. However, you should explain why this analysis is appropriate in light of the findings of Sergeev et al. (Sci Trans Med 2013, reference 20) who find that the "individual-level association between HIV and drug-resistant forms of TB is dynamic, and therefore, cross-sectional studies that do not report a positive individual-level association will not provide assurance that HIV does not exacerbate the burden of resistant TB in the community."

*Also, while the analysis includes HIV and TB/MDR-TB, it's not clear on how it specifically fits in with the rest of paper or if it could be cut without loss of any key information.*

*You report in Figure 5, the correlation between global patterns of MDR-TB, TB and HIV prevalence. In the Methods you refer to the World Health Organization for the underlying data. This reference should be extended as it now only states "World Health Organization". Data from several countries are known to have limitations in the sense that the prevalence of HIV or TB may not always been known or collected in a rigorous manner. The prevalence of a particular disease can also vary within a single country. Use of these data can therefore also result in biased estimates, as estimates for HIV and TB may be derived from different sites in a single country.*

This entire analysis needs to be either strengthened or simply removed from the paper.

We appreciate our conclusion may seem in variance with the sentence copied from the Abstract of the paper by Sergeev et al. However, we feel it is important to put their statement in its context. The Sergeev et al. paper deals with the dynamics of drug resistant TB (DRTB) over the course of the HIV pandemic in a single setting (Zimbabwe). They focus on a quantity defined as the “ratio of the proportion of TB cases that are drug resistant among HIV-seropositive individuals to the proportion of TB cases that are drug resistant among HIV-seronegative individuals”.

The paper highlights the interesting behavior over time of this ratio, but this is not a quantity we consider. The paper does not explore the effect of different rates of incidence/prevalence of HIV and does not make predictions for other settings than the situation in Zimbabwe they model. The sentence from their Abstract makes an interesting point about the behavior of this ratio in their simulations and states that depending if it is recorded early or late in the same epidemic, it can take different values. However, we do not think the statement can be expanded to warn against cross-comparisons devoid of individual HIV/TB status over multiple settings all recorded at the same time, late in the HIV pandemic (in our case 2013).

The only figure from the Sergeev paper that can be directly compared to the quantities we consider in our correlations is their Figure 3, which reports the simulated incidence of TB and DRTB in the presence or absence of HIV. Interestingly the simulated patterns they report are actually in line with our global correlations. Except in the very early stages of the epidemic (before ~1990), populations experiencing an HIV epidemic also suffer a higher burden of TB (our panel 5B), a higher burden of MDR-TB (our panel 5C), but the ratio of MDR-TB/ TB remains constant (our panel 5D).

The Sergeev paper also makes a couple of seemingly reasonable assumptions that are likely important determinants of the modelled individual-level rates of drug resistant TB by HIV status as well as overall resistant/susceptible ratio over time. Their model assumes a lower fitness threshold in HIV+ individuals, which necessarily leads to resistant strains (that have lower fitness) being more likely to cause disease in HIV positives (although the model includes compensatory mechanisms that reduce the fitness cost over time in resistant bacteria).

Less important for individual-level relationships but probably an important shaper of the modelled overall rates of drug resistant TB is the parametrization of transmissibility in the model, which assumes that HIV+ TB patients transmit TB less efficiently. It is difficult for us to assess this in detail, but the combination of these two assumptions (1. HIV+ patients are more prone to become infected and contagious with drug resistant TB. 2. HIV+ transmit TB less efficiently than HIV-), that are expected to have opposing effects on the modelled association between HIV prevalence and DR-TB/TB ratio, might explain why the overall MDR/DR/DS ratio is constant irrespective of the presence of HIV (Figure 3 in the Sergeev paper).

However, as our goal was to assess the effect of HIV on transmission, resistance and latency using real data, we did not parametrize HIV+ and HIV- people separately, as this would defeat the point of our analyses. There is however (surprisingly) little support in our results for different fitness thresholds depending on HIV status (both resistance load and estimated *Mtb* mutation rates are identical between HIV classes) and also little support for a negative association between HIV co-infection and *Mtb* transmissibility.

The strength of WHO epidemiological data is their availability for most of the world. Even if country-wide summaries can mask some heterogeneities at a smaller scale, we believe the high level of replication (large number of countries) allows capturing robust qualitative global patterns. However, we recognize that there are differences in methodologies between countries and that data from low income countries are likely to be less reliable (which could lead to biases).

To ensure the robustness of the correlations we report, we have now tested whether GDP per capita or public health expenditure per capita might color our results. Limiting the analyses to countries with the highest GDP or health expenditure did indeed increase the correlations between HIV prevalence, TB prevalence and MDR_TB prevalence but did never lead to a significant correlation between HIV prevalence and the ratio of MDR-TB over susceptible TB prevalence (new Figure 5 and new Figure 5—figure supplement 1).

We appreciate these correlations are not directly linked to the rest of the analyses in the paper. In order to confirm our findings in multiple settings, it will be important to replicate our analyses to other outbreaks or epidemics. However, we feel that the inclusion of Figure 5 offers some support for our findings being general throughout different settings. That said, we recognize that the best place for opening up the debate on the generality of our results is probably not the Results section, and we have now moved the figure and the associated text to the Discussion.

Finally, we agree that the rationale, justification and methodological details were inappropriately detailed in the previous draft and wish to apologize for this oversight. We are now far more explicit on the origin of the data and the analysis and have attached the source data to Figure 5 ([Supplementary-material SD3-data]). The changes to the manuscript are listed below”

Inclusion of a fourth panel in Figure 5, showing the correlation between TB prevalence and MDR-TB prevalence;Figure 5 moved to Discussion and a re-write of the section in the text pertaining to Figure 5 (Discussion, fifth paragraph);Description of the data and analyses in Methods (subsection “Collection of data on global HIV, TB and MDR-TB burden”);Inclusion of a supplementary figure with the same correlations than in Figure 5 but limited to the countries with top 50% GDP per capita.

*4) More explanation is required for key numbers reported in their paper.*

In particular, in the Results, you state: "Based on the available data we considered that the sequenced outbreak isolates represented about 35% of the total number of individuals belonging to the outbreak." You should explain how you came up with the figure of 35%. What are you assuming about the fraction of TB cases that are diagnosed and recorded? How, and over what time scale, are you defining the "outbreak" since reactivation TB cases may have been infected decades prior to disease onset?

We have added an explanation for our estimates of sampling density in the second paragraph of the subsection “Impact of HIV co-infection on *Mtb* mutation rates and resistance development”. We now refer to one third rather than 35% which is actually more accurate (252 sequenced genomes out of an estimated total of 778). The outbreak start is assumed to be 1992, when the outbreak was detected (Ritacco et al. J infect Dis 1997). We have however shown that the MRCA of the large six-drug resistant clade of the outbreak had evolved already by around 1979, but the strain seems to have circulated in very small numbers prior to 1992, when it caused an outbreak at the Muniz hospital in Buenos Aires (Eldholm et al. 2015 Nat Commun).

*Again, in the Results, you state: "This left 13 resistance mutations that evolved with high probability during therapy in 11 patients (Table 2). Seven of the patients were HIV negative and four were positive. A χ2 analysis revealed a statistically significant overrepresentation of acquired resistance in HIV-negative cases relative to positives (p = 0.027). This finding strongly suggests that HIV was not a driver of Mtb drug resistance within the outbreak." Given that you indicate that these results strongly back up a key part of your findings, it is important that more information be provided. Specifically, it is unclear exactly which numbers were being compared in the test (e.g. was the null hypothesis that there should be a 50:50 split between HIV-positives and negatives, or was it directly comparing the fraction of mutations found in HIV-infected vs. uninfected individuals [though this would not seem to give the p-value reported]). Also, a Fisher's exact test, not a chi-squared test, would likely be required given that apparently some of the cell sizes were very small.*

We would like to thank the reviewer for identifying this rather weak piece of statistics. The expected numbers are based on the fraction of HIV positives and negatives in the sampled outbreak. We have now redone the statistics using the Fisher’s exact test, which indeed shows that the difference is not significant:

obsexpneg95pos48p=0.24

We would however like to stress that we were aware that the numbers are too small to draw strong conclusions and that we originally stated (as you acknowledge above) that these results strongly suggested that HIV co-infection was not a driver of *Mtb* resistance, rather than the much stronger alternative, namely that HIV co-infection could somehow be said to be “protective” against resistance development. We still feel that the results back up a key finding, as the distributions do not deviate from the 0-hypothesis that there is no difference between the two groups, i.e. HIV is not a direct driver of resistance emergence in the study setting. We have updated the subsection “Impact of HIV co-infection on *Mtb* mutation rates and resistance development” (last paragraph) accordingly and moderated our statement.